# Cardiovascular Risk Assessment among Farmers in French Guiana in 2018—A Screening Program

**DOI:** 10.3390/ijerph20021262

**Published:** 2023-01-10

**Authors:** Amélie Martinot, Antoine Adenis, Paul Brousse, Yoland Govindin, Cyril Rousseau, Nadia Thomas, Mathieu Nacher, Timothee Bonifay

**Affiliations:** 1Centre Hospitalier de Cayenne, Cayenne 97300, French Guiana; 2Mutualité Sociale Agricole de Guyane, Cayenne 97300, French Guiana; 3Centre Régional de Coordination des Dépistages des Cancers, Cayenne 97300, French Guiana

**Keywords:** farmers, cardiovascular risk, public health, French Guiana, cross-sectional study

## Abstract

Context: There is a general health decline among farmers and the leading cause of death in this population remains cardiovascular (CV) diseases. The situation is similar in the Guianese general population, with a preoccupying increase in CV diseases. However, there are no data on farmers’ health. Methods: A cross-sectional study analyzed data from the “Novembre Vert” action conducted in 2018 in French Guiana. Beneficiaries and farmers affiliated to the Mutualité Sociale Agricole who completed the survey were included. The objective was to assess their CV risk. Results: 603 farmers were included. The sex-ratio was 1.6 and the median age was 52. Over 70% of the participants had a Body Mass Index ≥ 25, with a greater risk of obesity in the female population. High blood pressure (HBP) affected 53.1% of farmers and 80.1% were diagnosed during screening. About 13.5% had diabetes. Overall, 27% of participants were at high or very high CV risk. CV risk was 3 times greater in men. Conclusion: HBP (53.1%), obesity (30.3%) and diabetes (13.5%) prevalence are particularly worrying and underline the importance of policies to reduce cardiovascular morbimortality among farmers.

## 1. Introduction

The reorganization of the agricultural world initiated in the 1950s has multiplied pressures and risks among farmers. Initially better than in the general population, their health indicators have now declined. Farmers are subject to profession-specific health problems—musculoskeletal disorders, psycho-social risks, cancers, reproductive diseases, sensory deficits [1]—but also to global health problems, including cardiovascular diseases, the leading cause of death in the French farming population [2].

Until today, few studies have focused on the agricultural world in French Guiana. However, this overseas territory located in the Amazon between Suriname and Brazil has specific problems: Its ethnically diverse population is experiencing very strong demographic growth (fertility rate of 3.82 children per woman [3]) and a fragile socio-economic situation with half of the population living below the national poverty line [4]. This results in specific health problems: a higher prevalence of high blood pressure (40.6% vs. 31.3% in mainland France), diabetes (7.7% vs. 5.7%), overweight and obesity (52% vs. 41%), kidney failure and strokes [5,6,7,8]. As a result, cardiovascular diseases are still the leading cause of death.

French Guiana also has local farming specificities with small farms, food-producing or family-run agriculture, low level of professionalism, etc. Furthermore, it is also the only French territory recording a constant increase in the number of farms. But farmers do not benefit from medical follow-up as part of their professional activity. In order to improve the access to healthcare the Mutualité Sociale Agricole (MSA) conducts in French Guiana a health survey every four years since 2014 called “Novembre Vert”. The MSA manages the specific health insurance for agricultural professions and provides its occupational medicine services. The main objective of this study was to describe the cardiovascular risk of Guianese farmers in 2018.

## 2. Materials and Methods

### 2.1. Study Design and Population

A cross-sectional study was carried out by the MSA from 1 to 30 November 2018 in French Guiana. From September 2018, all active farmers affiliated to the MSA were contacted by post, telephone and radio to propose and promote the operation. An appointment date was set in the farmer’s choice town. A reminder call was made ten days before and the day before the appointment. A mobile medical team moved around the entire territory and offered consultations in 17 of the 22 communes of French Guiana. The farmers domiciled in French Guiana and affiliated to the MSA who completed the survey were included in the study. Participants completed a questionnaire then received parameter’s recording and a medical consultation. During the visit, anthropometric measurements and parameters were taken (weight, height, two blood pressure readings (BP), Fasting Plasma Glucose (FPG), urine dipstick, vision tests and audiogram).

### 2.2. Variables

A cardiovascular risk score was developed based on the recommendations of the European Society of Cardiology and the French Society of Endocrinology [9,10]. The following variables each corresponded to a risk factor (RF): (1) age over 60 years for women and over 50 years for men. (2) High blood pressure (HBP), defined either by the patient’s knowledge of a history of high blood pressure or by the mean of two blood pressure measurements ≥ 140/90 mmHg (meanSBP/meanDBP). HBP was classified, according to the French Society of Cardiology definitions, into three stages or grades: grade 1 or mild (mean SBP between 140 and 159 mmHg and/or mean DBP between 90 and 99 mmHg); grade 2 or moderate (mean SBP between 160 and 179 mmHg and/or mean DBP between 100 and 109 mmHg) and grade 3 or severe (mean SBP ≥ 180 and/or mean DBP ≥ 110). (3) Diabetes was defined either by a history or a discovery of diabetes, having a capillary FPG greater than 1.26 g/L (7 mmol/L) or a capillary postprandial plasma glucose measurement equal or greater than 2 g/L (11 mmol/L). (4) Body Mass Index (BMI). The World Health Organization classification was used to classify weight categories: thinness (BMI < 18.5 kg/m^2^), normal weight (BMI between 18.5 and 24 kg/m^2^), overweight (BMI between 25 and 29 kg/m^2^) and obesity (BMI ≥ 30 kg/m^2^). Obesity was the risk factor retained for CV score calculation. (5) Daily smoking and (6) Daily alcohol consumption.

The score took into account the grade of high blood pressure and the greater weight of diabetes in terms of CV risk [9,10]. The score was created to determine five risk profiles: (1) no CV risk; (2) low CV risk: “no HBP, 1 or 2 other RFs (diabetes excluded)” or “grade 1 HBP and no other RF”; (3) moderate CV risk: “no HBP, ≥3 RFs (diabetes excluded)” or “diabetes with no other RF” or “grade 1 HBP and one or two other RFs (diabetes excluded)” or “grade 2 HBP with no other RF”; (4) high CV risk: “HBP any grade and diabetes without other RF” or “HBP grade 3 without other RF” or “HBP grade 1 and ≥3 RFs (diabetes excluded)” or “HBP grade 2 and one or two other RFs (diabetes excluded)” or “no HBP, diabetes and 1 or 2 other RFs”; (5) very high CV risk: “HBP any grade and diabetes and other RF”, “HBP grade 2 and ≥3 RFs”, “HBP grade 3 and ≥1 RF”, “Diabetes and ≥3 RFs”.

### 2.3. Satistical Analysis

The database was processed using the Excel program. Statistical analyses were performed with STATA 12 software (STATA©, College Station, TX, USA) with a first-species risk of 5%. Comparisons between groups were first performed by bivariate analyses using χ^2^ tests, Student’s *t*-test, non-parametric Rank Sum tests or Fisher’s exact test for expected values < 5. Associations were then measured by calculating Odds Ratios (binary variables) or by logistic regression (ordinal variables) after harmonization of relative frequencies and distributions. A multivariate analysis of gender-adjusted BMI was performed.

### 2.4. Ethical and Legal Aspects

Oral and written consent were obtained for each participant. The study protocol was approved by the French Commission Nationale de l’Informatique et des Libertés (CNIL), complied with the MR004 Methodology Reference and was submitted to the Health Data Hub (authorization n°2223474). All investigators were trained to ensure reliability and harmonization of data collection.

## 3. Results

### 3.1. Participants

Among the 2041 MSA-affiliated farmers in 2018, 603 (29.5%) answered both the self-questionnaire and the medical consultation (Figure 1). The male:female sex-ratio was 1.6 and the median age was 52 years. Of all the participants, few had a professional qualification (9%, n = 54/603) and it concerned twice as many men (OR = 2.05, CI95% [1.04–4.26], *p* = 0.027) (Table 1).

Regarding Table 1, the farms were mostly small-scale crops (small surfaces and family owned). The declared use of chemical products concerned 50.9% of farmers (n = 307). On the contrary, 2.8% reported an organic agriculture management. Pesticide use was higher in the West and East (*p* = 0.012) and for crops (*p* < 0.001). Nearly 75% (n = 229/307) of farmers had attended the mandatory training course dedicated to the proper use of pesticides (Certyphyto).

### 3.2. Cardiovascular Risk

The most frequent risk factors were: high blood pressure and obesity in women, and age and high blood pressure in men (Table 2). Old age, smoking, alcohol consumption and unbalanced diet were less frequent in women than in men. The median BMI was 27.5 and 30.3% of farmers (n = 175/577) were obese with an excess risk in the female population (OR = 2.02, CI95% [1.38–3.0], *p* < 0.01) (Figure 2). Obese individuals were twice as likely to have high blood pressure (OR = 1.98; 95% CI [1.37–2.9]; *p* < 0.01) or diabetes (OR = 1.96; 95% CI [1.15–3.35]; *p* = 0.01). In all, 53.1% (n = 307/578) of the farmers were hypertensive, and 19.9% (n = 115/577) of BP measurements exceeded 160/100 mmHg; among these moderate and severe high blood pressures, 79.1% were new screenings. Finally, with no difference according to residence area or sex, blood pressure remained unbalanced in 70% of known hypertensive cases (n = 42/60). In total, 13.5% (n = 74/550) of the farmers had diabetes. Its prevalence increased with age (*p* < 0.01) and was lower in the West (OR = 0.52, CI95% [0.28–0.96], *p* = 0.035). Finally, with no difference according to residence area or sex, blood glucose levels remained high in 23.6% (n = 10/55) of known diabetics. The absence of CV risk concerned 21% (n = 122) of the farmers and on the contrary, 27% (n = 156) were at high or very high CV risk. All categories combined, CV risk was 3 times more frequent in men (OR = 2.96, CI95% [1.92–4.56], *p* < 0.01) and increased with age (*p* < 0.001).

### 3.3. Professional Qualification

Having received an agricultural course was a protective factor for diabetes (OR = 0.13, CI95% [0.003–0.80], *p* = 0.02), high and very high cardiovascular risk (OR = 0.22, CI95% [0.050–0.55], *p* < 0.01) and appeared to protect against high blood pressure (*p* = 0.056).

## 4. Discussion

The aim of this study was to assess the cardiovascular profile of 603 farmers throughout French Guiana. Cardiometabolic diseases are over-represented in the Overseas Departments and Territories (DROMs) and in French Guiana compared to mainland France [7], but our results were worse than expected (Table 3). They were in favor of high and very high cardiovascular risk for a large proportion of participants (27.0%), with a particularly worrying prevalence of high blood pressure (53.1%), obesity (30.3%) and diabetes (13.5%). There was also a significant proportion of unknown cases of hypertensive disease (80.1%, n = 246) and HBP ≥ 160/100 mmHg (19.1%, n = 115). These results were consistent with the literature (Table 3). High blood pressure increased with age, low educational level and affected more men [5,6,11]. Obesity increased the risk of high blood pressure and diabetes [12,13] and was over-represented in the female population [14,15]. These data also emphasized the importance of the combination of obesity and high blood pressure, particularly in women, in the onset of cardiovascular risk [5,12,16,17].

The main cardiometabolic risk factors in the general and agricultural populations were, as expected, obesity, BPH, and diabetes. Studies in the French West Indies identified obesity as the first modifiable factor linked to high blood pressure after aging [16,18]. The increase in obesity, as well as BPH and diabetes, has been explained by: (1) Westernization and sedentary lifestyle, which are more frequent in French Guiana (25% vs. 22% in mainland France) [7]. (2) Changes in eating habits: among farmers of this study, 43.8% did not eat fruit and vegetables every day. (3) Poor socio-economic conditions, which are recognized as a major determinant of overweight and ultimately of cardiovascular morbidity and mortality [9,10]. In French Guiana, there is a high level of precariousness (50% of people living below the poverty line) and a lower average level of education. It should be noted that women are more affected by these inequalities, which partly explains their increased risk of obesity. (4) At last, the important African and Asian ancestry of the Guianese population exposes them to an increased risk of high blood pressure and diabetes [9,19].

The present study had a number of limitations. The study did not consider either the retired or inactive farmers, nor those who did not fulfill the criteria for MSA affiliation. Yet, the most common form of agriculture in French Guiana is a familial and self-sufficiency farming. This represents 75% of farms [20] and is rarely declared. This type of practice does not allow MSA affiliation. Consequently, this study is not intended to be representative of the entire agricultural population of French Guiana. From a clinical point of view, due to the lack of confirmation during a second appointment, high blood pressure may have been over-diagnosed during the survey. On the contrary, the prevalence of diabetes was possibly underestimated and should be interpreted with caution given the low number of fasting plasma glucose measurements collected (n = 108). This work concerned 10.1% of the farm owners counted during the last agricultural census in French Guiana (2010) [21], yet few Guianese studies cover the entire territory and Novembre Vert is the only campaign that makes the collection of farmers’ medical data possible. In addition, there was little missing data (less than 5% per variable).

Novembre Vert campaigns, specific to French Guiana, give access to occupational medicine for hundreds of farmers. They are of great interest for public health and the study of risks related to agricultural professions as well. Given the size of the territory and its resources, these initiatives are to be congratulated, encouraged and made durable. The promotion of professional qualification and access to rights should be enhanced as it is strongly linked to cardiometabolic diseases and professional risks. Finally, a cohort study such as Agrican [2], set up in mainland France since 2005, would help answer several questions, particularly concerning the diagnosis of diseases associated with pesticides use or agricultural activity. Work-related illnesses are poorly admitted and frequently occur after the end of the activity: musculoskeletal disorders (MSDs) are the main cause followed by psychological suffering [5]. Other most studied diseases among farmers are related to pesticides. Multiple myeloma, non-Hodgkin’s lymphoma, chronic lymphocytic leukemia and Parkinson’s disease are recognized as occupational diseases for pesticide exposure ≥ 10 years. Other diseases with a strong presumption are prostate cancer, certain childhood cancers and congenital malformations. Alzheimer’s disease, cognitive deterioration and fertility disorders are also strongly suspected. Finally, melanoma and lip cancer are also more frequent but linked to sun exposure [1,22,23]. In French Guiana, the possibility of buying non-EU products in border countries exacerbates the lack of knowledge about pesticides use. It raises important issues and emphasizes that these work-related illnesses should be monitored in French Guiana.

## 5. Conclusions

In conclusion, this study highlights the importance of cardiovascular risk factors in the French Guianese agricultural population, notably the high prevalence of high blood pressure (53.1%), obesity (30.3%) and diabetes (13.5%). It remains essential to strengthen prevention, screening and treatment actions for French Guianese agricultural workers. These results also highlight the importance of regional public health policies aimed to reduce cardiovascular morbidity and mortality. In the end, it seems important to take a more global interest in this specific population.

## Figures and Tables

**Figure 1 ijerph-20-01262-f001:**
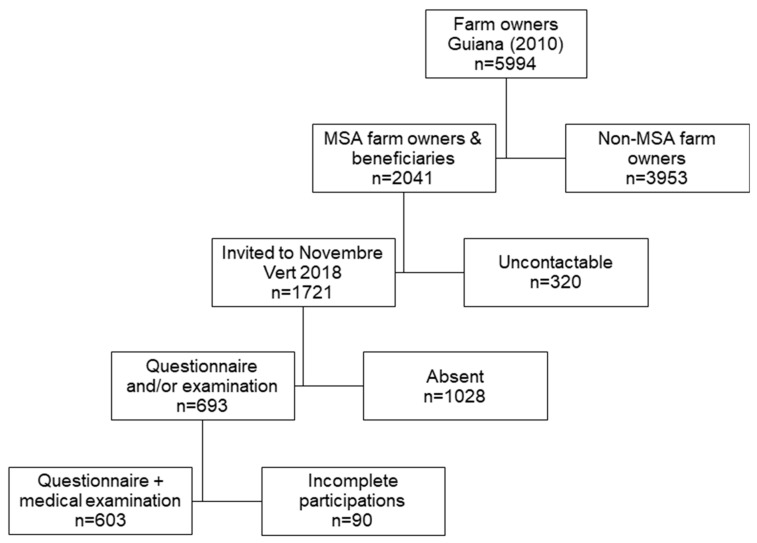
Flow chart of participants included in the study (MSA: Mutualité Sociale Agricole).

**Figure 2 ijerph-20-01262-f002:**
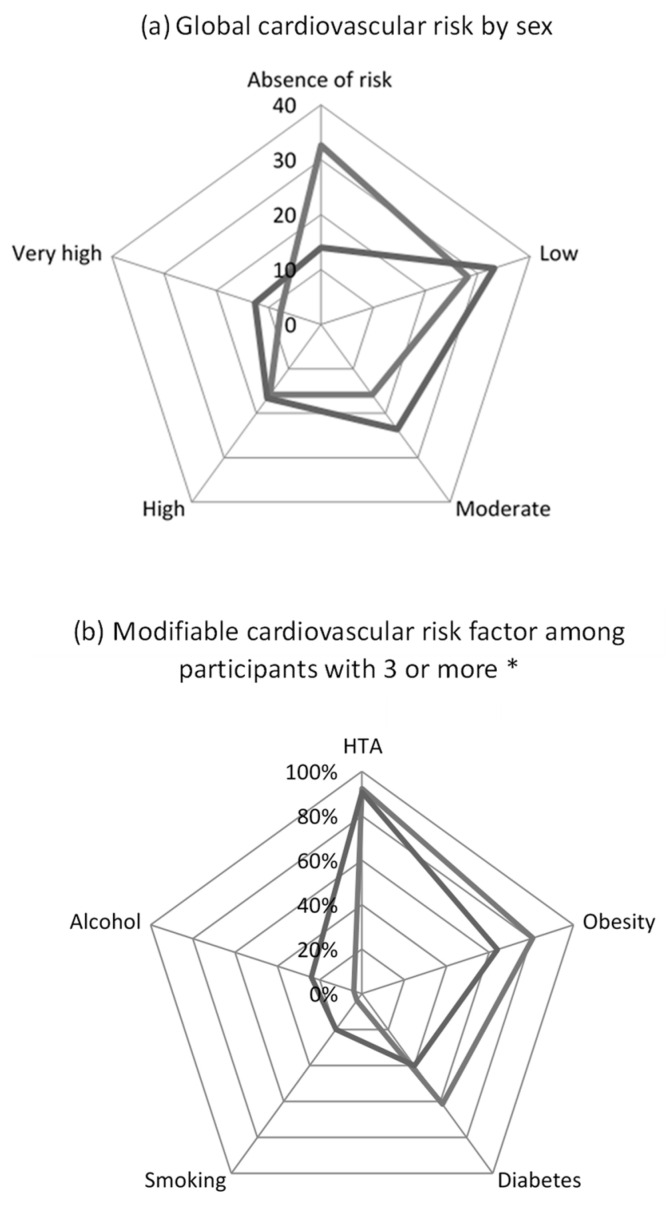
Modifiable cardiovascular risk factors among Novembre Vert 2018 participants with three or more risk factors by sex. * Among: age, HBP, obesity, diabetes, daily alcohol consumption and smoking (HBP = High blood pressure; RF = risk factor). Legend (**a**) in dark grey women (n = 221) and light grey men (n = 356), (**b**) in dark grey women (n = 26) and light grey men (n = 75).

**Table 1 ijerph-20-01262-t001:** Sociodemographic and farming characteristics of the 603 participants in Novembre Vert 2018.

	Total Participants (N = 603)		
	n	(%)	95% CI
Sociodemographic characteristics			
Women	229	(38.0)	34.1–41.9
Age, median (IQR), years *	52	(43–59)	
Age < 40 years	113	(18.7)	15.6–21.9
40–60 years	348	(57.7)	53.8–61.7
≥60 years	142	(23.6)	20.2–27.0
Household members ≥ 5 **¤**	241	(40)	36.1–44.0
Farm owner spouse	97	(16.1)	13.2–19.0
Town residence:			
Eastern	33	(5.5)	3.7–7.3
Savannas	90	(14.9)	12.1–17.8
Western	209	(34.7)	30.9–38.5
Coastal	271	(44.9)	41.0–48.9
Underwent farming course	54	(9.0)	6.7–11.2
Farming characteristics			
Livestock farming	61	(10.1)	7.7–12.5
Culture	468	(77.6)	74.3–81.0
Combined livestock/culture	74	(12.3)	9.7–14.9
Mechanized	265	(44)	40.0–47.9
Insured	263	(43.6)	39.7–47.6
Area, median (IQR), Ha *	5	(2–12)	
UAA ≤ 2 Ha	155	(25.8)	22.3–29.3
UAA 2 to 20 Ha	376	(62.5)	58.6–66.3
Workers number, median (IQR) *	2	(1–2)	
Organic farming	17	(2.8)	1.5–4.1
Pesticides use	307	(50.9)	46.9–54.9

* data presented as median (IQR); **¤** N = 602; UAA: Utilized agriculture area; Ha: hectare.

**Table 2 ijerph-20-01262-t002:** Cardiovascular risk factors of the Novembre Vert 2018 participants by sex.

Cardiovascular Risk Factors	Women		Men		OR [95% CI]	*p*
	N = 229	(%)	N = 374	(%)		
Age (W ≥ 60 y and M ≥ 50 y) *	42	(18.3)	224	(59.9)	6.65 [4.42–10.1]	<0.001
Daily smoking	4	(1.8)	47	(12.6)	8.09 [2.89–31.26]	<0.001
Daily alcohol	3	(1.3)	27	(7.2)	5.86 [1.77–30.5]	0.001
Obesity (BMI ≥ 30)	87/219	(39.7)	88/358	(24.6)	0.50 [0.34–0.72]	<0.001
Overweight [25–30]	77	(35.2)	156	(43.6)	1.46 [1.01–2.01]	0.046
Obesity ≥ 40	10	(4.6)	4	(1.1)	0.24 [0.05–0.83]	0.009
HBP ≥ 140/90	98/221	(44.3)	190/356	(53.4)	1.44 [1.01–2.04]	0.035
HBP history	27/79	(34.2)	34/185	(18.4)	0.43 [0.23–0.83]	0.005
HBP new screenings	79/194	(40.7)	167/323	(51.7)	1.56 [1.07–2.27]	0.016
HBP ≥ 180/110	7	(3.2)	25	(7.0)	2.31 [1.00–5.31]	0.049
Diabetes ^$^	29/210	(13.8)	45/340	(13.2)	-	-
History ^$$^	27/228	(11.8)	37/372	(10.0)	-	-
Pathological measurements ^$$$^	7/208	(3,4)	16/333	(4.1)	-	-
New screenings ^$$$$^	2/183	(1.1)	8/303	(2.6)	-	-
Moderate, high and very high cardiovascular risk	87/221	(39.4)	188/356	(61.7)	1.72 [1.21–2.46]	0.002

*p* is specified when significant. OR: Odd ratio, 95CI: 95% confidence interval, IQR: interquartile range, BMI: Body mass index, HBP: High blood pressure. * Age risk was 60 yo for women and 50 yo for men. HBP: n = 577 measurements. BMI: n = 577 measurements. Diabetes: ^$^ n = 550 blood sugar measurements and/or diabetes history; ^$$^ n = 600 medical history responses; ^$$$^ n = 541 blood sugar measurements, elevated when >1.26 g/dL after fasting or ≥2 g/dL postprandial; ^$$$$^ n = 486 non-diabetics blood sugar measurements. Cardiovascular risk score applied to 577 participants.

**Table 3 ijerph-20-01262-t003:** Cardiovascular risk factors comparison between French Guiana (general population and previous Novembre Vert) and mainland France populations.

	Novembre Vert 18	Novembre Vert 14	Baromètre Santé Guyane ^3,4^	Mainland France ^1,2^
Population	Farmers	Farmers	General	General
Year	2018	2014	2014	2014, 2015
Place	Guiana	Guiana	Guiana	Mainland
Size	603	732	2015	15,635; 2270
Age, mean (years)	52.0 (43–59)	50.0 (±10.4)	-	-
Age > 75 years	0.8	-	2	9
Daily smoking (%)	8.5	11.2	12	28
Daily alcohol (%)	5.0	7.3	4.8	9.7
Overweight (%)	40.4	40.3	34	29, 32
Obesity (%)	30.3	25.6	18	12, 17
HBP (%)	53.1	44.8 *	17.9, 40.6	31.3
Diabetes (%)	13.5	17.1 **	7.7	4.7, 5.7

^1^ Baromètre santé 2014 [7]; ^2^ ESTEBAN Study 2015; ^3^ Podium study by Attalah and Al 2008 [11]; ^4^ Observatoire Régional de la Santé study in Cayenne 2016 [5]; * HBP defined by Blood Pressure ≥ 140/90 mmHg; ** among the 110 fasting blood sugar measurements.

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
