# Peer review of "Cardiovascular Risk Assessment among Farmers in French Guiana in 2018—A Screening Program"

_ijerph, 2023, doi:10.3390/ijerph20021262_

Round 1

Reviewer 1 Report

 Cardiovascular risk assessment among farmers in French Guiana in 2018-a screening program.” Is a retrospective cross-sectional study to investigate the farmer’s risk of cardiovascular event in French Guiana.

It is a local study, however, I think it is important for the farmer’s healthcare in developing country.

Comments

1 Only 603 of the 2041 were finally recruited. It is feared that the results will be biased.

2 P2 line60-62, Did the medical team measure farmer’s blood pressure and FPG just one time? Was the Hypertension and/or Diabetes diagnosed only one point measurement? Blood pressure was measured twice at one point. FPG was measured just one time at one point. It may be difficult to make a correct diagnosis after a single consultation and examination?

3 P4 line120-126, “Agricultural characteristic” What does this paragraph mean?

4 Developing countries are usually considered to have lower rates of lifestyle diseases than Europe. Is it right to assume that the higher than expected incidence of hypertension and diabetes in this study was due to the westernization of diets and corresponding lower levels of education? If that is what the author is claiming, I think a more detailed discussion is needed.

Reviewer 2 Report

Abstract: You state the following: “The sex-ratio was 1.6.” This provides little information for the reader- was this the female:male or male:female ratio? Please clarify.

Please clarify how the sample size was determined. How many active farmers in French Guinana are not affiliated with the MSA? In what way would the exclusion of these farmers have affected the results of the study?

The study is characterized as both “retrospective” and “cross-sectional.” Please clarify why the study is characterized as “retrospective,” as it appears that all outcomes were based on cross-sectional evaluations of blood pressure, weight, height, etc.

There are numerous spelling and grammatical issues throughout the manuscript, and it would benefit from thorough editing via a professional English language editing service.

Reviewer 3 Report

1.Through the cardiovascular risk screening and assessment of farmers in the French Guiana, Having identified high rates of HBP, obesity and diabetes among local farmers. With this basic data, under the concept of One World, One Health, public health authorities should pay attention to this specific population, take corresponding preventive and therapeutic actions, and improve the health level of local farmers. 

2. There is an extra ") "between the first and second line and between the third and fourth line at the bottom of Table 1;There may be an error in line 4 from the bottom "UAA 2 à 20Ha".

Author Response

Response to Reviewer 3 Comments

Thank you for your remarks, indeed one of the objectives of the study is to alert the public health authorities,

The corrections on the table have been made.